# OpenReview forum: "Test-Time Backdoor Attacks on Multimodal Large Language Models"
_TMLR — Rejected by TMLR_

### Review · Reviewer_gsdi · 2025-10-28

**Summary Of Contributions:**

This paper introduces AnyDoor, a novel test-time backdoor attack targeting deployed Multimodal Large Language Models (MLLMs), such as LLaVA-1.5 and InstructBLIP, without requiring access to their training data or parameters. The attack's core mechanism exploits the MLLM's multimodal nature by decoupling the attack across modalities: an adversarial visual perturbation sets up the backdoor, and a textual trigger prompt activates the harmful output. The paper validates that this technique is highly effective, but also complex to optimize that may limit its practical use.

**Key Strengths**

- The paper successfully transitions the traditional backdoor concept from a training-time vulnerability to a deployment-time threat. This immediately raises the bar for MLLM safety and monitoring.
- The method is highly effective due to the division of labor: using the high-capacity visual modality for setup and the high-timeliness textual modality for activation.
- The attack's generalizability is thoroughly established by testing its effectiveness across four diverse and popular MLLM architectures (LLaVA-1.5, MiniGPT-4, InstructBLIP, and BLIP-2).

**Key Weaknesses**

- The attack relies on balancing two competing goals—attack efficacy and benign utility—which requires careful tuning of highly sensitive hyper-parameters (like the loss weights $w_1$ and $w_2$, perturbation budgets, and ensemble sizes). This necessitates significant, model-specific, and resource-intensive trial-and-error, which limits the practical use of the method.
- The attack's success is measured by two fundamentally different evaluation strategies ("With Trigger": Exact Match/Contain for efficacy; "Without Trigger": BLEU/ROUGE for utility). The inherent challenge of simultaneously optimizing for a perfect target match (Efficacy) while preserving high linguistic quality (Utility) reveals a major practical hurdle in achieving a robust and stealthy attack.
- The attack's reliability diminishes when the target output is a complex, structured response rather than a simple, short malicious text string.

**Audience:**

Yes

**Audience Explanation:**

The TMLR audience may be interested in the strategy developed in this paper on MMLM test-time attack, as well as the algorithm itself.

**Claims And Evidence:**

Yes

**Claims Explanation:**

The claims that AnyDoor is an effective test-time backdoor attack are supported by convincing evidence. The methodology is clearly defined through mathematical formulations and an explicit algorithm. Furthermore, the extensive ablation studies demonstrate that the attack is robust across various perturbation types and trigger variations, confirming the widespread nature of the vulnerability.

**Requested Changes:**

A further discussion on how to reduce the computational cost as well as make the algorithm easy to optimize would be great.

---

> ### Author Response · Authors · 2025-12-11
> **Rebuttal by Authors**
>
> Thank you for your supportive review and suggestions. Below we respond to the comments in **Weaknesses (W)**.
>
> ---
>
> ***W1: The attack relies on balancing two competing goals—attack efficacy and benign utility—which requires careful tuning of highly sensitive hyper-parameters.***
>
> We agree that balancing the two competing objectives—maximizing attack success (controlled by $w\_1$) and minimizing utility loss (controlled by $w\_2$),—is a primary concern in multi-objective attacks. However, our ablation studies demonstrate that we have incorporated mechanisms that make the optimization process more stable and less sensitive than it might appear:
>
> - **Dynamic Weight Strategy**: We investigated a **dynamic weight strategy** ($w\_1=\\lambda$, $w\_2=1−\\lambda$) that automatically balances the two competing objectives. This strategy significantly improves ExactMatch accuracy, BLEU@4, and ROUGE_L scores compared to using fixed weights. This practical approach **greatly lowers the barrier to deploying the attack** by eliminating the need for exhaustive manual hyper-parameter tuning.
>
> - **Robust Attack Strategy**: The **Border Attack** strategy proved to be **the most effective** strategy across all three datasets evaluated, providing a stable and practical default for perturbation placement.
>
> - **Enhanced Generalization**: Increasing the **ensemble sample size** further stabilizes performance, as an expanded sample size of 200 on VQAv2 yielded the highest BLEU@4 and ROUGE_L scores ($\\textrm{\\color{blue}Table 3}$ (Page 8)), confirming that larger ensembles correlate with higher benign accuracy.
>
> These findings confirm that while the conceptual problem is challenging, AnyDoor incorporates mechanisms that make its practical optimization surprisingly robust,.
>
> ---
>
> ***W2: The inherent challenge of simultaneously optimizing for a perfect target match (Efficacy) while preserving high linguistic quality (Utility) reveals a major practical hurdle in achieving a robust and stealthy attack.***
>
> Preserving high linguistic quality for normal queries is essential for the stealthiness of any backdoor attack. We addressed this practical hurdle through explicit optimization and refined evaluation protocols:
>
> - **Explicit Optimization for Utility**: Our target attack condition mandates that for benign inputs, the perturbed output must match the clean output. The $w\_2$ term in our loss function ($\\textrm{\\color{blue}Eq. 2}$ (Page 5)) directly controls this minimization of utility loss.
>
> - **Adoption of LAVE Metric**: To accurately assess the linguistic quality and semantic consistency of the MLLM's open-ended generation, we report Benign Accuracy computed via the LLM-Assisted VQA Evaluation (LAVE) protocol, as standard string matching metrics tend to underestimate performance.
>
> - **Experimental Validation of Utility**: Our control experiments ($\\textrm{\\color{blue}Table 2}$ (Page 7)) validate that the attack preserves benign utility in most practical scenarios. The Clean baseline achieves 98.5% Accuracy, and the Trigger-only setting maintains 94.4% Accuracy. This demonstrates that the $w\_2$ constraint effectively limits the degradation of the model's linguistic performance for benign queries.
>
> ---
>
> ***W3: The attack's reliability diminishes when the target output is a complex, structured response rather than a simple, short malicious text string.***
>
> We acknowledge this observation, as our ablation studies show that the complexity of the target output significantly impacts attack reliability.
>
> Attacks targeting concise target phrases result in consistently high ExactMatch scores across datasets. These simple malicious strings (like short system commands) represent a significant real-world threat.
>
> We confirm that when targeting intricate function call strings, the attack’s success rate diminishes significantly or fails entirely. We attribute this drop in reliability mainly to the inadequacy of the default 500-PGD step optimization process for such complex, high-structure outputs. We suggest that increasing the number of optimization iterations is necessary to improve effectiveness against complex targets, highlighting the importance of parameter calibration for varying target complexities.
>
> This trade-off emphasizes that while AnyDoor is highly effective against specific, predefined short malicious outputs, achieving robust success against complex, structured outputs requires deeper optimization efforts.

---

### Review · Reviewer_oaFY · 2025-10-31

**Summary Of Contributions:**

The submission presents a AnyDoor, a test-time backdoor attack aimed at multi-modal LLMs.
AnyDoor relies on an universal adversarial attack on the visual component, used as a backdoor setup, and on a textual trigger to elicit the desired behavior.
The authors present a series of experiments showing the relative efficacy of the backdoor attack.

**Audience:**

Yes

**Audience Explanation:**

LLM attacks is an extremely important research direction, which is definitely of interest to a broad section of the community.

**Broader Impact Concerns:**

Given the significant risk of misuse associated with the presented attacks, the authors should definitely add a Broader Impact statement.

**Claims And Evidence:**

No

**Claims Explanation:**

I believe that the submission has two main weaknesses: one experimental, the other conceptual.

Conceptually, the authors stress the distinction between adversarial attacks and their approach by pointing to the distinction between the time the backdoor is setup (when the attack is first deployed), and when the trigger is leveraged.
However, in practice, as far as I understand, in order to induce the desired behaviour, both the textual trigger and the visual attack need to be present in the current input. Effectively, to me, this means that the current test input is an attack, and that, the notation of the authors $t_{set}=t_{act}$. I do not really see much of an advantage in the fact that the visual attack can be started before the prompt modification.
This is particularly important, as this distinction is basically the only technical contribution presented by the authors, setting the presented method apart from standard universal adversarial attacks.

Concerning the experiments: they lack baselines. At the very least, they should compare the metrics for the "without trigger" case with the behavior of the network without any visual attack. It would also be interesting to see the difference between the efficacy of a standard universal attack alone to induce the desired behavior (compared with the "with trigger" case).
But it would also be interesting to compare against other types of backdoor attacks (not at test-time), even though I appreciate this is a different context.

**Requested Changes:**

- I believe the authors should provide concrete examples on contexts in which the use of their method is preferable in practice to the use of a simple adversarial attack to induce the desired behavior. I am not particularly convinced by the provided example on the adversarial sticker and then the prompt. Other examples should be needed.
- The experiments should have at the very least baselines of unattacked networks and networks with standard universal adversarial attacks, as per my comments above.

---

> ### Author Response · Authors · 2025-12-11
> **Rebuttal by Authors**
>
> Thank you for your supportive review and suggestions. Below we respond to the comments in **Weaknesses (W)**.
>
> ---
>
> ***W1: The authors should provide concrete examples on contexts in which the use of their method is preferable in practice to the use of a simple adversarial attack to induce the desired behavior.***
>
> We respectfully clarify that the key distinction in our work is not about when the attacker locally computes the perturbation (which can be done for any attack), but about **when the model is exposed to the malicious components**.
>
> In a standard universal adversarial attack (UAP), **setup and activation are inseparable**: the perturbation must be present at the exact moment the attacker wants to induce malicious behavior, and it immediately affects the system’s output once applied.
>
> In contrast, AnyDoor defines a **test-time backdoor architecture**:
>
> - **Setup phase**: The attacker injects the universal visual perturbation long before the attack is activated. The perturbation remains fully dormant under all benign prompts.
>
> - **Activation phase**: Much later, a non-adversarial, human-readable text prefix selectively activates the malicious output.
>
> This separation provides two practical advantages that UAPs fundamentally cannot achieve:
>
> - **Stealth**: The adversarial visual perturbation does not degrade the robot/agent’s performance until the trigger is given. This avoids early detection.
>
> - **Control**: Attackers can safely attach the perturbation at a time when the system is unwatched (setup), and activate the attack later via text alone.
>
> As illustrated in $\\textrm{\\color{blue}Figure 2}$ (Page 3), a physical UAP requires placing the patch precisely at the attack moment, risking exposure. AnyDoor enables placement earlier—when safe—and activation only at a sensitive moment.
>
> This two-phase separation is a defining property of classical backdoor attacks, and our work shows that **test-time backdoors can exist without poisoning training data**, expanding the scope of multimodal security research.
>
> ---
>
> ***W2: The experiments lack baselines.***
>
> Thank you for your constructive comments. We have incorporated extensive ablation studies, particularly in $\\textrm{\\color{blue}Table 2}$ (Page 7), to provide the requested baselines. We detail below how our results address your concerns, particularly highlighting the unique setting of AnyDoor as a Test-Time Backdoor Attack (TTBA) compared to standard adversarial and traditional poisoning attacks.
>
> We report the performance of the unattacked network, referred to as the "Clean" setting, in Table 2. This establishes the model's normal operational performance without any visual attack:
>
> - Clean Baseline: The network achieves 98.5% Accuracy and 0.0% ASR (Attack Success Rate).
>
> The metrics for our controlled settings (e.g., Trigger-only at 94.4% Accuracy and 0.0% ASR) are directly compared against this clean baseline to confirm that the method generally preserves the model's utility for benign queries. Even when the UAP is applied without the textual trigger, the ASR (Exact Match) is only 13.5%, significantly lower than the 89.5% achieved when the trigger is present.
>
> A direct comparison with a standard UAA is crucial, but it requires acknowledging the fundamental difference in the attack objective: AnyDoor is designed as a **Test-Time Backdoor Attack**, which requires the attack to be **conditional**. Our primary objective is to create a conditional harmful effect while preserving high benign utility when the trigger is absent. This is achieved by strategically decoupling the burden of setup and activation to different modalities.
>
> As shown in $\\textrm{\\color{blue}Table 2}$ (Page 7), the results confirm that, due to the benign utility (i.e., stealth) constraint, our perturbation is engineered not to cause unconditional, universal model failure; its harmful characteristics are suppressed. The significant increase in ASR when the textual trigger is introduced validates that our UAP successfully acts as the backdoor **setup**, requiring the textual trigger as the activation key.
>
> Furthermore, AnyDoor fundamentally differs from standard universal adversarial attacks at the threat model level, rendering direct quantitative comparison inappropriate: AnyDoor is a multimodal attack that strategically decouples the tasks of setting up and activating the harmful effect.
>
> Standard universal adversarial attacks require both setup and activation to occur at the exact same timing ($t_{set}=t_{act}$), imposing the burden on a single modality. AnyDoor's multimodal decomposition allows for the separation of these timings ($t_{set}$ and $t_{act}$) during the test phase, which allows AnyDoor to focus on the test-time backdoor attack threat model, achieving **conditional, multimodal, and decoupled** attacks.

---

> > ### Comment · Reviewer_oaFY · 2025-12-16
> >
> > I thank the authors for their response.
> >
> > Unfortunately, my concerns remain:
> > - I am still not convinced by the validity of the stressed distinction between conditional and unconditional attacks as, in practice, the visual attack is always present.
> > - Thank you for Table 2, but such an important evaluation should have been carried out systematically on all benchmarks; it seems like the UAP alone has a non-negligible impact on performance, and could hence be spotted as such.
> > - It would be important to compare against the efficacy of UAPs used alone to trigger malicious behaviour (that is, as standard adversarial attacks).

---

### Review · Reviewer_CVGH · 2025-11-29

**Summary Of Contributions:**

This paper proposes a white-box, input-perturbation-only adversarial attack on MLLMs that learns a visual pixel perturbation to elicit a fixed harmful output prefix only when a specific text prefix is present.

Rather than optimizing a "universal" adversarial image patch that always forces the model to produce the target (harmful) text regardless of the text prompt, the attack is gated by a small text trigger used in the loss. Conceptually, this yields a multi-trigger attack and simulates a backdoor: a trained visual trigger/patch that does the heavy lifting, and a lightweight text trigger whose co-presence is required to activate the attack.

In the motivating example of an MLLM-based robot, this means that one can place adversarial stickers to breach the underlying MLLM, but defer the activation of the attack until a fixed text input is present.

The strength of the paper includes:
- Clear and novel settings of a text-gated image patch attack, nicely set across modalities.
- Practically motivated by the example of an MLLM-based robot
- Good investigation of the attack's effectiveness across settings of trigger text selection, visual patch size (budget), visual patch forms, trigger texts, target texts, and patch robustness. Minimal overstatement throughout.

The weakness of the paper includes (for details, see requested changes):
- The writing needs to be improved: terminology usage is unsatisfactory, and some explanations come too late.
- The choice of the presented evaluation metric is questionable: e.g., the lack of accuracy evaluation for visual QA datasets.
- The assumption on attacker's resources is somewhat strong (whitebox) while the achieved effect (fixed output prefix) is less strong than tasks like jailbreaking. I am not sure how making model output "I want to destroy the world" is useful to anyone.
- Some baseline is missing, and the models used are a little outdated.
- Insufficient demonstration of the motivation behind the modality delegation.

**Audience:**

Yes

**Audience Explanation:**

I believe the setup and findings of the proposed attack are generalizable to some TMLR's audience.

**Broader Impact Concerns:**

A common practice of adversarial attack works is to gate the published artifacts and code with an ethical and responsible usage agreement. The author should promise the same.

**Claims And Evidence:**

Yes

**Claims Explanation:**

- Experiments are comprehensive: the paper investigated an attack across three visual QA datasets. The attack was constructed with three different strategies under different perturbation budgets, "ensemble" (training) size, and loss terms weights.
- The authors also demonstrated the attack's robustness against some common image transformations.
- However, the reported metrics are not yet fully convincing.

**Requested Changes:**

## Significant changes:
- The authors should report the accuracy of the VQA dataset, both with and without triggers. In fact, BLEU@4 and Rougue_L seem so correlated that reporting either is more than enough. If the authors choose not to report the accuracy, a convincing reason must be provided.
- The authors should report the `exact match` for all without trigger setting (aka false positive rate). This tells us "how conditional is this conditional attack".
- The authors should report a baseline of just blacking out the corner/border/selected pixels, as the visual patch might actually block some view and disable the model from answering some questions from the VQA image. If the authors choose not to report any baselines, a convincing reason must be provided.
- The author should clearly establish that with only the trigger or only the perturbation or neither, the asr is consistently 0% (or some low value).

## Other changes:
- The discussion of the visual perturbation being a medium with "better capacity" is confusing. I do not understand what "capacity" is at first, nor am I convinced that it is the case that it is easier to attack a model with images than using texts. In fact, the "Huang et al., 2023" mentioned in the last line of the first paragraph of A.2 seems to share a comparable setup, but the attack was entirely done in text space.
- "Timeliness" is such a confusing term that only becomes clear to me after the introduction of the MLLM-based robot with a camera sticker and the description in 3.1. Before that, I was so confused why you need to time anything in a chatbot-style system until I realized a robot might be continuously recording and running a model. Consider removing "timeliness" entirely before 3.1 (especially in the abstract), and maybe use other words like "conditional" / "composite" / "thresholded" / "backdoor-style" or "deferred/delayed-activation attack" if needed. Maybe "perturbation's non-interference for the triggerless text prompts" or something alike
    - > "Activating harmful effects, on the other hand, requires strong manipulating timeliness to ensure that the harmful effects are triggered at the appropriate time."

        In addition to the fact that "requires strong manipulating timeliness" is not a grammartical sentence, I was very lost at what is an  "appropriate time". Move the robot example as early as needed.
- The definition of "harmful effect" was unclear to me until very late. In fact, I initially thought the attack was jailbreaking until much later. I was still confused when I was reading the top part of page 5. Furthermore, considering the upper right of figure 1, the desired effect in this paper is more of a fixed term $A$ rather than $A_n$. Furthermore, the fact that the attack is only eliciting fixed text provides less utility than jailbreaking, especially for "SUDO" kind of wording.
- Figure 1 the authors squeezed the vast field of "Adversarial Attack" into a single LLM attack scheme and imply that their work is not an adversarial attack... Not standard in the field. Consider using any other names -- "Adversarial, Universal Perturbation" or something. Please fix this for all.
- Figure 1 by the current drawing I would never realize the authors mean (only) the border is taped with the sticker. Also the border used in the experiment is much thicker than illustrated.
- In sec 2, consider swapping the order of "multimodal adversarial attacks" with "universal adversarial attack". Apparently, the latter comes first.
- "....require modifying model weights or structures" by "structure", do you mean "architecture"?
- "...fooling multiple images at the same time" I suppose you mean fooling the models rather than the images?
- "Li et al., 2022d" should be in related work too, at least in the full one.
- Please specify dataset sizes. Also, specify if you used the train split for the ensemble and the test split for reporting in the table.
- Table 2 comes up on page 6, when most things in the table were not yet described. Consider swapping it with page 7 or placing it later? Table 1 is early too. Many tables and plots in this paper are too far from their corresponding text.
- > it is commonly observed that textual input has limited capacity to be manipulated but can be easily intervened upon at any time (such as giving instructions to a robot) (Zou et al., 2023).

    Similar to a previously requested change, this requires more evidence. How can it be easily intervened? What do you mean by giving instructions in this case? Do you mean instruct the model to discard GCG-style inputs? Does that work?
- in 3.2 "AnyDoor adaptively assigns each modality" is an overstatement. The work only trains on visual patches and triggers with text prefix.
- in 3.3 please discuss which previous work(s) made or inspired you to choose the three attacking strategies.
- Perhaps I am not very familiar with these MLLM, but what exactly is the image input size used? What does $\epsilon = */255$ mean? Why 255?
- in 4.2 consider using "Loss term weights" instead of "Loss weight". Word choices like this can use more clarity.
- For the random targets, does outputting any harmful prefix in the pool count as success?
- Table 7 has two ExactMatch columns
- Explanation of Table 7 and Table 9's missing entries and attacking strategy rows is necessary. Table 9 really needs a baseline of just noise patches or blank patches to situate the reported numbers.

## Suggestions:
- The author could report a skyline of unconditional visual attack and compare the attack success rate and BLEU/ROUGUE to demonstrate the comparably small deviation when visual perturbation is present but the text trigger is not.
- The author might wish to go further and test "sudo" attack success rate for "SUDO"-trained only results (in addition to currently trained "sudo/SUDO"-trained results), and unlike settings.
- The proposed and experimented defenses to attacks demonstrated in this paper are extremely simple. Some more sophisticated defense, comparable to the text perplexity filter that defends GCG-style suffix, might be needed.

---

> ### Author Response · Authors · 2025-12-11
> **Rebuttal by Authors [1/4]**
>
> Thank you for your supportive review and suggestions. Below we respond to the comments in **Significant Changes (SC)** and **Other Changes (OC)**.
>
> ---
>
> ***SC1: The authors should report the accuracy of the VQA dataset, both with and without triggers.***
>
> Thank you for highlighting the importance of baseline experiments. Following your suggestions, we conducted additional experiments under different conditions to evaluate the accuracy of the VQA dataset, ensuring a more comprehensive assessment. Specifically, we adopted the LLM-Assisted VQA Evaluation (LAVE) [1] to measure benign accuracy, which is robust for open-ended generation.
>
> As shown in  $\\textrm{\\color{blue}Table 2}$ (Page 7), the model maintains high utility in the Trigger-only setting (94.4% accuracy) and the Black-patch setting (98.0% accuracy), comparable to the Clean baseline (98.5%). Even under our attack setting (UAP-patch without trigger), the model retains a decent accuracy of 85.0%, demonstrating that our method preserves the model's functionality on benign queries.
>
> [1] Improving Automatic VQA Evaluation Using Large Language Models
>
> ---
>
> ***SC2: The authors should report the exact match for all without trigger setting (aka false positive rate).***
>
> We reported the Attack Success Rate (ASR) for the "without trigger" settings, which serves as the false positive rate. As shown in  $\\textrm{\\color{blue}Table 2}$ (Page 7), when the UAP is applied without the textual trigger, the ASR (Exact Match) is only 13.5%, significantly lower than the 89.5% achieved when the trigger is present. This confirms that AnyDoor is highly conditional: the attack is only fully activated when both the visual setup and textual key align.
>
> ---
>
> ***SC3: The authors should report a baseline of just blacking out the corner/border/selected pixels.***
>
> Thanks for your suggestion. We introduced a "Black-patch" baseline (masking the border pixels) to isolate the effect of visual occlusion. The results in $\\textrm{\\color{blue}Table 2}$ (Page 7) show that the Black-patch achieves 98.0% benign accuracy, which is nearly identical to the Clean baseline (98.5%). This confirms that the performance drop in our attack (to 85.0%) is driven by the adversarial nature of the perturbation, not by the visual obstruction of the border itself.
>
> ---
>
> ***SC4: The author should clearly establish that with only the trigger or only the perturbation or neither, the asr is consistently 0% (or some low value).***
>
> $\\textrm{\\color{blue}Table 2}$ (Page 7) clearly establishes the necessity of our method for successful attacks. The ASR is consistently 0.0% for the Clean, Trigger-only, Black-patch, and Black-patch + Trigger settings. In contrast, our full attack (UAP-patch + Trigger) achieves a high ASR of 89.5%. This verifies that neither the trigger alone nor a simple black border can activate the backdoor; the attack requires the specific universal adversarial perturbation.

---

> ### Author Response · Authors · 2025-12-11
> **Rebuttal by Authors [2/4]**
>
> ---
>
> ***OC1&OC2: The discussion of the visual perturbation being a medium with "better capacity" and "Timeliness" is confusing.***
>
> We thank the reviewer for these insightful comments. We agree that the terms "capacity" and "timeliness" were confusing and that the motivation could be better presented. We have unified our response as these points are interconnected regarding the mechanism's logic.
>
> We have removed the confusing terms "capacity" and "timeliness" from the Abstract, Introduction, and Section 3. Instead, we now use more precise descriptions consistent with the manuscript's logic:
>
> - **Visual Modality for Setup**: We clarify that visual inputs (continuous pixel space) offer a "richer embedding space" (i.e., high optimization degrees of freedom) [2], making them ideal for embedding complex, dormant adversarial perturbations (setup) without affecting benign performance.
>
> - **Textual Modality for Activation**: We now describe the text modality as offering superior "controllability" (i.e., flexible on-demand activation), highlighting its role in acting as a precise key to trigger the attack exactly when intended.
>
> As suggested, we have moved the "Robot with a Camera Sticker" example (originally Figure 2) to the last paragraph of $\\textrm{\\color{blue}Introduction}$ (Page 2). This immediately provides an intuitive grounding: the sticker acts as the static "setup" (hard to change instantly), while the user's text command acts as the dynamic "activation" (triggered at any appropriate time).
>
> Regarding Text Attacks[3], we acknowledge that purely text-based attacks are effective. However, our argument is not that image attacks are intrinsically "easier," but that decoupling the attack exploits the unique strengths of MLLMs: shifting the heavy optimization burden to the visual encoder (setup) allows the text encoder to process natural, trigger-based instructions (activation) efficiently at test-time.
>
> [2] Improving Automatic VQA Evaluation Using Large Language Models \
> [3] Composite Backdoor Attacks Against Large Language Models
>
> ---
>
> ***OC3: The definition of "harmful effect" was unclear until very late.***
>
> We apologize for the confusion regarding the definition of "harmful effect." We clarify that AnyDoor is a Backdoor Attack, not a Jailbreak attack. Specifically, Jailbreaking aims to bypass safety filters to answer varied malicious queries (e.g., "How to build a bomb?"). AnyDoor (Backdoor) aims to force the model to output a specific, pre-defined target string (A_harm) whenever a trigger is present. As correctly observed by the reviewer, our goal is indeed a fixed term A_harm (as defined in $\\textrm{\\color{blue}Eq. 1}$ (Page 4)).
>
> We respectfully argue that eliciting fixed text has immense utility, especially for MLLM Agents and Tool-use scenarios. In these contexts, the model's output often triggers executable actions (e.g., function calls, robotic controls). As demonstrated in $\\textrm{\\color{blue}Table 1}$ (Page 7) ("Target Function Call") and $\\textrm{\\color{blue}Table 17}$ (Page 26) (Target Pool), AnyDoor can inject malicious code or commands as the fixed target, such as *rm -rf*, *DROP TABLE users*, or *{"func_name": "transfer_money"}*. This allows an adversary to hijack the agent to execute a specific dangerous action (e.g., forcing a robot to "drop" an object or a system to "delete" files) simply by showing a trigger, which constitutes a severe security threat distinct from jailbreaking.
>
> ---
>
> ***OC4: Figure 1 the authors squeezed the vast field of "Adversarial Attack" into a single LLM attack scheme.***
>
> We appreciate the reviewer’s concern. $\\textrm{\\color{blue}Figure 1}$ (Page 2) was never meant to represent the entire adversarial-attack field; rather, it served as a conceptual illustration contrasting unconditional universal perturbations and our proposed gated test-time backdoor structure (conditional).  We updated the caption to explicitly state the differences.
>
> ---
>
> ***OC5: Figure 1 by the current drawing I would never realize the authors mean (only) the border is taped with the sticker.***
>
> We thank the reviewer for careful observation. We assume this comment refers to $\\textrm{\\color{blue}Figure 2}$ (Page 3), as Figure 1 describes the timeline.
>
> In our specific implementation (Border Attack), the "sticker" is conceptually a transparent film with a noise pattern only on the rim. The center region remains clean. In our experiments, as shown in $\\textrm{\\color{blue}Table 1}$ (Page 7), we use a border width of b=6 pixels for a 336×336 image resolution, which is the standard setting for LLaVA-1.5. This is intentionally designed to be subtle (stealthy). To avoid confusion, we have updated the caption to explicitly state the physical adversarial sticker.

---

> ### Author Response · Authors · 2025-12-11
> **Rebuttal by Authors [3/4]**
>
> ---
>
> ***OC6-OC9&OC16: Refinements on Organization, Terminology, and Citations.***
>
> We thank the reviewer for the meticulous reading and valuable suggestions. We have incorporated all the requested adjustments:
>
> - We swapped the order of "multimodal adversarial attacks" with "universal adversarial attack".
> - We replaced “...require modifying model weights or structures” with the more precise term “model architecture.”
> - The phrase “fooling multiple images” has been corrected to “fooling multiple models.”
> - Added Li et al., 2022d to the full related work section (Section A.2) as suggested.
> - Replaced "Loss weights" with "Loss term weights".
>
> ---
>
> ***OC10: Please specify dataset sizes. Also, specify if you used the train split for the ensemble and the test split for reporting in the table.***
>
> Thanks for your suggestion. We have revised Section 4 ("Datasets" and "Attacking strategies") to explicitly detail our protocol. For VQAv2 and SVIT, we strictly utilize the Training Split for ensemble optimization and the Validation Split for evaluation. For the synthetic DALL-E 3, we generated two distinct, non-overlapping pools for these purposes. About dataset sizes, as specified in the "Sample Size" column of $\\textrm{\\color{blue}Table 1}$ (Page 7), we use a default of 40 samples for optimization (we also ablate sizes from 40 to 200 in $\\textrm{\\color{blue}Table 3}$ (Page 8). For all reported results, we used a fixed set of 200 samples randomly drawn from the validation split. We revise it for clarity in the paper revision (Section 4).
>
> ---
>
> ***OC11: Many tables and plots in this paper are too far from their corresponding text.***
>
> We thank the reviewer for pointing out the layout issues. We have extensively reorganized the manuscript.
>
> We have moved the main results table (formerly Table 2) to Section 4.1 $\\textrm{\\color{blue}Table 1}$ (Page 7), placing it after the "Attacking strategies" and "Evaluation metrics" paragraphs. And, we have moved the trigger-target ablation table (formerly Table 1) to Section 4.2 $\\textrm{\\color{blue}Table 5}$ (Page 8).
>
> ---
>
> ***OC12: How can textual input be easily intervened in?***
>
> We apologize for the confusion caused by the phrasing "easily intervened upon." We clarify that we do not mean a defense mechanism (e.g., instructing the model to discard GCG inputs). Instead, "intervened" here was intended to describe the flexible, real-time interactivity of the textual modality from a user's perspective.
>
> In our context (e.g., the robot scenario in $\\textrm{\\color{blue}Figure 2}$ (Page 3)), the visual backdoor (sticker) is physically static and hard to modify instantly ("setup"). In contrast, the textual input serves as a dynamic control channel where a user can issue instructions (triggers) on demand and remotely at any specific moment ("activation"). This capability to "interact" with the model via text at any time is what differentiates it from the static visual setup. We have revised the sentence in $\\textrm{\\color{blue}Section 3.1}$ to replace the confusing phrase with “flexible for real-time user interaction” to accurately reflect this motivation.
>
> ---
>
> ***OC13: In 3.2 "AnyDoor adaptively assigns each modality" is an overstatement.***
>
> We appreciate the reviewer for pointing out this overstatement. We have revised the sentence in Section 3.2 to be precise, by replacing "adaptively assigns" with “strategically leverages the distinct characteristics of modalities by assigning the visual modality to the 'setup' task and the textual modality to the 'activation' task.”
>
> ---
>
> ***OC14: In 3.3 please discuss which previous work(s) made or inspired you to choose the three attacking strategies.***
>
> Thank you for pointing this out. We chose the strategies to bridge theoretical baselines with physical realism:
>
> - **Border Attack**: Directly inspired by "Adversarial Framing"[4] and "Adversarial Camera Stickers"[5], simulating physical stickers on lenses.
> - **Corner Attack**: Adapted from standard Adversarial Patches [6] to minimize central occlusion.
> - **Pixel Attack**: Included as a standard theoretical baseline following [7].
>
> We have updated Section 3.3  to explicitly cite these works as the source of our designs.
>
> [4] Adversarial Framing for Image and Video Classification \
> [5] Adversarial Camera Stickers: A Physical Camera-based Attack on Deep Learning Systems \
> [6] Adversarial Patch \
> [7] Towards Deep Learning Models Resistant to Adversarial Attacks

---

> ### Author Response · Authors · 2025-12-11
> **Rebuttal by Authors [4/4]**
>
> ---
>
> ***OC15: What exactly is the image input size used? What does $\\epsilon$ = \*/255 mean? Why 255?***
>
> We appreciate the opportunity to clarify these details.
>
> In our main experiments, we evaluated the LLaVA-1.5 model, which integrates the Vicuna-7B and Vicuna-13B language models using a default input resolution of 336×336.
>
> For $\\epsilon$ = */255, this notation specifies the perturbation budget ($\epsilon$) used in the Pixel Attack strategy. This attack involves introducing adversarial perturbation across the entire image using an $l_{\infty}$ constraint. The budget limits the maximum amplitude of the perturbation. Our default setting for the Pixel Attack uses $\epsilon$=32/255.
>
> The value 255 serves as the normalization factor used to define the magnitude of the perturbation budget. The reason for choosing 255, it is standard practice in vision literature to reflect the maximum intensity value (0–255) of an 8-bit color channel.
>
> ---
>
> ***OC17: For the random targets, does outputting any harmful prefix in the pool count as success?***
>
> We emphasize that our evaluation is strictly target-specific. In the random pairing experiment, for each randomly selected pair ($Trigger_i$, $Target_i$), we count it as a success if and only if the model generates the specific assigned target $Target_i$ (measured by *ExactMatch* and *Contain* metrics).
>
> This strict criterion demonstrates that AnyDoor allows the attacker to inject a precise backdoor payload, rather than merely inducing generic toxic behavior. We have clarified this definition in Section 4.2.
>
> ---
>
> ***OC18: Table 7 has two ExactMatch columns***
>
> We apologize for the ambiguity. The two "ExactMatch" columns serve distinctly different purposes:
>
> - **Under "With Trigger"**: It measures Attack Success Rate, checking if the output matches the malicious target.
>
> - **Under "Without Trigger"**: It measures Benign Accuracy, checking if the output matches the ground truth. As noted in Section 4.3, we use ExactMatch for benign evaluation here because models like BLIP-2 and InstructBLIP produce very concise answers, making standard captioning metrics (BLEU/ROUGE) less effective.
>
> We have renamed the headers in  $\\textrm{\\color{blue}Table 8 (former Table 7)}$ (Page 9) to **"ASR (ExactMatch)"** and **"Benign Acc. (ExactMatch)"** and redrawn the table for improved clarity.
>
> ---
>
> ***OC19: Explanation of Table 7 and Table 9's missing entries and attacking strategy rows is necessary.***
>
> We thank the reviewer for pointing out the need for clearer baselines and explanations regarding missing entries. The missing entries ("-") are intentional and due to the distinct response behaviors of different MLLMs, as discussed in Section 4.3.
>
> - **Concise Models (e.g., BLIP-2, InstructBLIP)**: These models generate very short answers. Standard captioning metrics like BLEU/ROUGE are not applicable/reliable for such short texts, so we used ExactMatch and marked BLEU/ROUGE as "-".
> - **Verbose Models (e.g., MiniGPT-4)**: These generate detailed descriptions where ExactMatch is too strict for benign inputs. Thus, we used BLEU/ROUGE and marked ExactMatch as "-".
>
> We have updated the captions of $\\textrm{\\color{blue}Table 8 (former Table 7)}$ (Page 9)  and $\\textrm{\\color{blue}Table 10 (former Table 9)}$ (Page 11)  to explicitly clarify these metric choices.
>
> Also, we have added a "Random Noise" baseline (using Gaussian noise with the same budget for LLaVA) to $\\textrm{\\color{blue}Table 10 (former Table 9)}$ (Page 11)  in the revision. As shown in the revised table, adding random noise (with the same budget b=6) has a negligible impact, maintaining high similarity scores (BLEU@4: 73.6, ROUGE_L: 88.5). In contrast, our transferred Border Attack causes a significant drop (BLEU@4: 59.5, ROUGE_L: 81.5).

---

### Author Response · Authors · 2025-12-11
**Summary of Paper Revision**

We thank all reviewers for their constructive feedback, and we have responded to each reviewer individually. We have also uploaded a **Paper Revision** including additional results and illustrations:

- $\\textrm{\\color{blue}Figure 2}$ (Page 2): e 6): Incorporate a practical physical-world scenario explanation where an adversarial sticker is applied to a robot’s camera.
- $\\textrm{\\color{blue}Table 2}$ (Page 7): VQA accuracy and ASR(ExactMatch) of various control baseline experiments (Clean, Black-patch, UAP-patch, Trigger-only, etc.).
- $\\textrm{\\color{blue}Table 1}$ (Page 7): We move the main results table (formerly Table 2) to Section 4.1.
- $\\textrm{\\color{blue}Table 5}$ (Page 8): We relocate the trigger-target ablation experiment (formerly Table 1).
- We reorganize tables and plots for improved clarity.

---

### Decision · Action_Editor_zAVC · 2026-02-01

**Recommendation:** Reject

**Audience:**

Yes

**Audience Explanation:**

Backdoor attacks on multimodal language models are of large interest to the ML/security research community right now, especially as these models are getting increasingly deployed in real-world settings.

**Claims And Evidence:**

No

**Claims Explanation:**

The reviewers raised a couple key concerns, which I have independently validated are still present in the revised version of the paper.

- **Nonstandard and inclear definitions:** The draft defines standard words in nonstandard ways (e.g. not considering a "backdoor attack" as a type of "adversarial attack", referring to a natural language instruction as an "embedding space"). The draft also makes assumptions that the visual signal is coming from a continuous video stream while text commands come as sporadic inputs. These assumptions are required in order for "textual modality being highly controllable" to be true, but this is not sufficiently clearly stated in the paper. Finally, acronyms (such as "UAP") are used without being defined. After reading through the paper, I believe it falls slightly below the "claims supported by ... clear evidence" bar of TMLR.
- **Insufficient ablations and baseline comparisons:** Reviewer oaFY remains concerned that the baselines are insufficiently covered, with Table 2 comparing different attack settings on only one of the three datasets. I share this concern, that the 13.5% attack success rate without the complete methods should be explored more in order to meet the TMLR bar for "convincing evidence."

In summary, I do think the paper's topic is of interest to TMLR's audience and that both these issues can be fixed, but the fixes would constitute major revisions, not minor ones.

**Resubmission Of Major Revision:**

The authors may consider submitting a major revision at a later time.